# Obesity Is Indirectly Associated with Sudden Cardiac Arrest through Various Risk Factors

**DOI:** 10.3390/jcm12052068

**Published:** 2023-03-06

**Authors:** Yun Gi Kim, Joo Hee Jeong, Seung-Young Roh, Kyung-Do Han, Yun Young Choi, Kyongjin Min, Jaemin Shim, Jong-Il Choi, Young-Hoon Kim

**Affiliations:** 1Division of Cardiology, Department of Internal Medicine, Korea University College of Medicine, Korea University Anam Hospital, Seoul 02841, Republic of Korea; 2Division of Cardiology, Department of Internal Medicine, Korea University College of Medicine, Korea University Guro Hospital, Seoul 08308, Republic of Korea; 3Department of Statistics and Actuarial Science, Soongsil University, Seoul 06978, Republic of Korea; 4Division of Cardiology, Sanggye Paik Hospital, Inje University College of Medicine, Seoul 01757, Republic of Korea

**Keywords:** sudden cardiac arrest, body-mass index, waist circumference, central obesity

## Abstract

Although obesity is a well-established risk factor of cardiovascular event, the linkage between obesity and sudden cardiac arrest (SCA) is not fully understood. Based on a nationwide health insurance database, this study investigated the impact of body weight status, measured by body-mass index (BMI) and waist circumference, on the SCA risk. A total of 4,234,341 participants who underwent medical check-ups in 2009 were included, and the influence of risk factors (age, sex, social habits, and metabolic disorders) was analyzed. For 33,345,378 person-years follow-up, SCA occurred in 16,352 cases. The BMI resulted in a J-shaped association with SCA risk, in which the obese group (BMI ≥ 30) had a 20.8% increased risk of SCA compared with the normal body weight group (18.5 ≤ BMI < 23.0) (*p* < 0.001). Waist circumference showed a linear association with the risk of SCA, with a 2.69-fold increased risk of SCA in the highest waist circumference group compared with the lowest waist circumference group (*p* < 0.001). However, after adjustment of risk factors, neither BMI nor waist circumference was associated with the SCA risk. In conclusion, obesity is not independently associated with SCA risk based on the consideration of various confounders. Rather than confining the findings to obesity itself, comprehensive consideration of metabolic disorders as well as demographics and social habits might provide better understanding and prevention of SCA.

## 1. Introduction

Obesity, which is associated with various medical diseases such as hypertension, diabetes mellitus, dyslipidemia, and coronary artery disease, is a major health concern in developed countries [1,2,3,4]. The association between obesity and coronary artery disease is of concern since it can lead to myocardial infarction or sudden cardiac arrest (SCA) [5,6,7,8]. A prospective cohort study involving 1 million adults in the United States revealed that the risk of all-cause death and cardiovascular death was significantly increased in obese people [8]. Another study with 2.3 million adolescents from Israel with 40 years of follow-up demonstrated that obesity during adolescence was associated with significantly increased all-cause and cardiovascular mortality in adults [7]. A study from the Republic of Korea also found a significant association between obesity and all-cause mortality [9]. In these studies, body weight status such as normal weight, overweight, and obesity was defined using body-mass index (BMI) criteria, an indicator of general obesity [7,8,9].

Using BMI can have several limitations [10]. It cannot divide (i) lean body mass from fat and (ii) abdominal fat from the fat of other body sites [10,11]. Furthermore, abdominal obesity, which is not fully reflected in BMI, can have a better predictive value for medical diseases such as diabetes mellitus and myocardial infarction compared with general obesity [12,13].

Sudden cardiac arrest is a medical emergency that imposes a significant burden on both the victim and the society [14,15,16]. Although both high BMI and waist circumference are known to be associated with the increased risk of cardiovascular death, whether this association is a direct effect of obesity or the result of metabolic comorbidities frequently associated with obesity, such as hypertension, diabetes mellitus, and dyslipidemia, is debated. In addition, prior studies only adjusted a limited number of covariates such as age, sex, height, smoking status, alcohol consumed, educational level, and level of physical exercise, and the influence of other important covariates such as blood pressure, fasting blood glucose, dyslipidemia, and estimated glomerular filtration rate (eGFR) were not taken into account [7,8,17]. We aimed to evaluate the association between obesity measured as BMI and waist circumference and the risk of SCA under adjustment of various metabolic comorbidities using a large prospective cohort from the Korean National Health Insurance Service (K-NHIS) database.

## 2. Materials and Methods

### 2.1. K-NHIS Database

This study is a retrospective analysis based on the K-NHIS database, which represents the entire population of South Korea. The K-NHIS is the single and exclusive medical insurance system managed by the government which mandates virtually the entire Korean population to subscribe to. The system is paid for by a nationwide tax system; it also covers those who are not able to afford it, and guarantees basic health care services (citizens are all registered in the system and, therefore, there is less chance of selection bias). The K-NHIS database offers a prospective cohort of subscribed citizens with medical records and various medical measurements during national health check-ups. Therefore, medical data derived from the K-NHIS database are a valuable source for a range of medical research.

If the protocols of the study are approved by both the institutional review board and the official review committee of the K-NHIS (https://nhiss.nhis.or.kr/, accessed on 21 January 2022), researchers are permitted to utilize the K-NHIS database to perform medical research. The Institutional Review Board of Korea University Medicine Anam Hospital and official review committee of the K-NHIS approved this specific study (IRB No.: 2021AN0185). The requirement for written informed consent was waived by the Institutional Review Board of Korea University Medicine Anam Hospital. This study complied with the Declaration of Helsinki and the legal regulations of South Korea.

The K-NHIS provides a regular, biennial, nationwide health check-up to its subscribers. The national health check-up is free of charge for the subscribers as it is covered by the government tax system. During the health check-up, various medical measurements are taken that include height, body weight, waist circumference, blood pressure, serum creatinine, liver function tests, fasting blood glucose (FBG), lipid profile, smoking and alcohol habits, level of income, and physical activity. In the K-NHIS database, various diagnostic codes of the International Classification of Disease, 10th revision (ICD-10) such as hypertension, diabetes mellitus, or heart failure, and prescription history of drugs are recorded. The capability of utilizing these covariates is a distinguished feature of medical research studies based on the K-NHIS database [18,19].

### 2.2. Participants

In 2009, 66% of people who were meant to undergo the nationwide health check-up actually underwent the check-up. Among adult citizens who underwent nationwide health check-ups in 2009, 40% were randomly sampled and enrolled in this study. Exclusion criteria were participants who were younger than 20 years or those with a diagnosis of SCA prior to enrollment (day of 2009 health check-up). Data obtained from 1 January 2002 to 31 December 2008 were used to identify baseline demographics such as presence of hypertension and diabetes mellitus. Medical follow-up duration was between the day of the 2009 health check-up of each participant and 31 December 2018. No follow-up losses were present except for emigrations.

### 2.3. Primary Outcome

The primary outcome is the occurrence of SCA during the follow-up period (the day of the 2009 health check-up of each patient and 31 December 2018). The incidence of SCA was defined as event numbers per 1000 person-years of follow-up. Identification of SCA events was based on claims of the following ICD-10 codes: I46.0 (cardiac arrest with successful resuscitation); I46.1 (sudden cardiac arrest); I46.9 (cardiac arrest, cause unspecified); I49.0 (ventricular fibrillation and flutter); R96.0 (instantaneous death); and R96.1 (death occurring less than 24 h from onset of symptoms). According to the definition of SCA, only claims that occurred at emergency department visit were identified as SCA event, and claims during in-hospital admission were excluded.

In order to conform to the definition of SCA, any possible non-cardiac causes of sudden arrest were excluded from the primary outcome [20]. If participants had a prior diagnosis of cerebral hemorrhage, ischemic stroke, asphyxia, suffocation, drowning, gastrointestinal bleeding, sepsis, anaphylaxis, major trauma, hit by lightning, electric shock, or burn within six months of the diagnosis of SCA, the event was not counted as a primary outcome.

### 2.4. Definitions

The influence of waist circumference and BMI on risk of SCA was evaluated. Waist circumference was measured as the mid-point between the rib cage and the iliac crest. Waist circumference was classified into six stages: waist circumference < 80.0 (cm), 80.0 ≤ waist circumference < 85.0, 85.0 ≤ waist circumference < 90.0, 90.0 ≤ waist circumference < 95.0, 95.0 ≤ waist circumference < 100.0, and waist circumference ≥ 100.0 for males, and waist circumference < 75.0, 75.0 ≤ waist circumference < 80.0, 80.0 ≤ waist circumference < 85.0, 85.0 ≤ waist circumference < 90.0, 90.0 ≤ waist circumference < 95.0, and waist circumference ≥ 95.0 for females. Body-mass index was classified into five groups: low body weight (BMI < 18.5 [kg/m^2^]); normal body weight (18.5 ≤ BMI < 23.0); pre-obesity (23.0 ≤ BMI < 25.0); obesity class I (or mild obesity, 25.0 ≤ BMI < 30.0); and obesity class II-III (or moderate to severe obesity, BMI ≥ 30.0) [21,22].

Alcohol consumption status was defined as follows: (i) non-drinker, 0 g of alcohol per week; (ii) mild to moderate drinker, <210 g of alcohol per week; and (iii) heavy drinker, ≥210 g of alcohol per week.

For smoking status: (i) current smokers were defined as those who smoked ≥ 100 cigarettes in their lifetime and continued smoking within one month of the 2009 nationwide health check-up; (ii) ex-smokers were those who smoked ≥ 100 cigarettes in their lifetime, but had not smoked within one month of the 2009 nationwide health check-up; and (iii) never-smokers were those who smoked < 100 cigarettes in their lifetime.

Diabetes mellitus and hypertension were classified into three stages each: (i) non-diabetic (FBG < 100 mg/dL); (ii) impaired fasting glucose (IFG) (FBG 100–125 mg/dL); and (iii) diabetes mellitus (FBG ≥ 126 mg/dL or a prior claim of ICD-10 codes for diabetes mellitus) for diabetes mellitus, and (i) non-hypertension (systolic blood pressure [SBP] < 120 [mmHg] and diastolic blood pressure [DBP] < 80); (ii) pre-hypertension (either 120 ≤ SBP < 140 or 80 ≤ DBP < 90); and (iii) hypertension (either SBP ≥ 140, DBP ≥ 90, or a prior claim of ICD-10 codes for hypertension) for hypertension.

Estimated glomerular filtration rate (eGFR) was calculated based on measured creatinine level during the 2009 health check-up, and chronic kidney disease (CKD) was defined as eGFR < 60 mL/min/1.73 m^2^ based on the Modification of Diet in Renal Disease (MDRD) equation.

Defining regular physical activity was based on a self-questionnaire acquired during the 2009 health check-up: people who had one or more sessions in a week with high (such as running, climbing, intense bicycle activities) or moderate (such as walking fast, tennis, or moderate bicycle activities) physical activity. The quality of physical measurement and laboratory tests are guaranteed and legally certified by K-NHIS, and the robustness of the aforementioned definitions was validated in our prior studies [19,23,24,25,26,27].

### 2.5. Statistical Analysis

The categorical variables are presented as number and percentage, and the continuous variables are presented as mean and standard deviation, or median value with interquartile range. The Student’s t-test was used for comparison of continuous variables, and the Chi-square test or Fisher’s exact test was used for comparison of the categorical variables as indicated. The Cox proportional hazards model was used to calculate unadjusted and adjusted hazard ratios (HR) and 95% confidence intervals (CI). In addition to the un-adjusted model, five multivariate models were adopted: (i) multivariate model 1: adjusted for age and sex; (ii) multivariate model 2: adjusted for model 1 plus smoking, alcohol, regular exercise, and income; (iii) multivariate model 3: adjusted for model 2 plus hypertension, diabetes mellitus, and dyslipidemia; (iv) multivariate model 4: adjusted for model 2 plus hypertension, diabetes mellitus, dyslipidemia, and chronic kidney disease; and (v) multivariate model 5: adjusted for model 4 plus ɣ-GTP. All tests were two-tailed, and statistical significance was defined as *p* values ≤ 0.05. All statistical analyses were performed with SAS version 9.2 (SAS Institute, Cary, NC, USA).

## 3. Results

### 3.1. Study Population

A total of 4,234,341 participants were randomly sampled from participants that underwent 2009 nationwide health screening (Figure 1). People with prior diagnosis of SCA (n = 491) and with missing data (n = 177,427) were excluded from the study and 4,056,423 people were followed until December 2018. Sudden cardiac arrest occurred in 16,352 subjects among 33,345,378 person-years of follow-up, with an incidence of 0.490 (per 1000 person-years). The flow of the study is summarized in Figure 1. Significant differences in the baseline demographics between people who did and did not experience SCA are summarized in Appendix A: people with SCA were older and had higher prevalence of male sex, current smokers, hypertension, diabetes mellitus, dyslipidemia, and CKD [25]. The baseline demographics according to BMI status demonstrated a significant difference across all parameters such as age, sex, smoking and alcohol consumption status, regular exercise, income level, hypertension, diabetes mellitus, dyslipidemia, CKD, and ɣ-glutamyl transferase (ɣ-GTP) (Table 1). A similar pattern of difference in the baseline demographics was observed according to waist circumference, which is described in Table 2.

### 3.2. BMI and SCA

Body weight status measured by BMI was significantly associated with the risk of SCA for both men and women (Table 3, Figure 2a). Moderate to severe obesity (BMI ≥ 30) had 20.8% increased rate of SCA compared with normal weight (18.5 ≤ BMI < 23) (95% CI = 1.12–1.31; *p* < 0.001: Table 3). After adjustment of the influence of age and sex, the increased rate of SCA in moderate to severe obese people was elevated to 35.5% from 20.8% (95% CI = 1.25–1.47; *p* < 0.001: Table 3). The relative risk of SCA in moderate to severe obese people was 38.8% higher after further adjusting for smoking and alcohol consumption status, regular exercise, and income level (95% CI = 1.28–1.50; *p* < 0.001: Table 3). However, the association between BMI and the risk of SCA was lost after adjusting for the influence of hypertension, diabetes mellitus, dyslipidemia, and CKD (HR = 1.05; 95% CI = 0.96–1.13; *p* = 0.286: Table 3, Figure 2a). Furthermore, people with pre-obesity (23 ≤ BMI < 25) and mild obesity (25 ≤ BMI < 30) showed significantly lower risk of SCA compared with people with normal body weight (18.5 ≤ BMI < 23) after multivariate adjustment (HR = 0.80 and 0.79, respectively; *p* < 0.001 for both: Table 3). The multivariate model further adjusting for ɣ-GTP showed similar results, reflecting no association between obesity and risk of SCA (HR = 0.94; 95% CI = 0.87–1.02; *p* = 0.130: Table 3, Figure 2a) and decreased risk of SCA in the pre-obesity and mild obesity groups (HR = 0.78 and 0.74, respectively; *p* < 0.001 for both: Table 3, Figure 2a).

### 3.3. Waist Circumference and SCA

Participants were classified into six groups according to waist circumference measured during their health check-up. Without adjustment of covariates, waist circumference showed a significant linear association with the risk of SCA with higher waist circumference associated with increased risk of SCA for both men and women (Table 3, Figure 2b). However, such association was significantly weakened after adjusting age, sex, smoking, alcohol, regular exercise, and income (Table 3, Figure 2b). After further adjusting the influence of metabolic disorders (hypertension, diabetes mellitus, dyslipidemia, and CKD), the highest waist circumference group no longer showed an increased rate of SCA (HR = 1.04; 95% CI = 0.96–1.12; *p* = 0.346). Compared with the reference group (<80 cm and 75 cm for men and women, respectively), middle-level waist circumference (between 80 cm and 100 cm for men, and 75 cm and 95 cm for women) was associated with lower risk of SCA (Table 3). Adjustment of ɣ-GTP further affected the association between waist circumference and the risk of SCA, with all other groups showing lower risk of SCA compared with the reference group (Table 3).

### 3.4. Obesity, Metabolic Syndrome and SCA

The association of obesity and SCA was further analyzed according to the presence of the classic metabolic syndromes—hypertension, diabetes mellitus, and dyslipidemia. Participants were divided into (i) those who had the three metabolic syndromes triad (hypertension, diabetes mellitus, and dyslipidemia) and (ii) those who had only one or two, or none of the metabolic syndrome triad Appendix A. Participants with the metabolic syndrome triad revealed a higher incidence of SCA across all subgroups. After adjusting covariates, participants with the metabolic syndrome triad did not show any significant association between obesity (measured with either BMI or waist circumference) and increased risk of SCA.

### 3.5. Multivariate Model

In a multivariate Cox-proportional-hazards model, age, sex, smoking status, alcohol consumption, regular exercise, low income, hypertension, diabetes mellitus, dyslipidemia, CKD, and ɣ-GTP were independently associated with SCA risk (Table 4). Influence on SCA based on the degree of hazard ratio was most prominent in age, sex, smoking, hypertension, diabetes mellitus, and CKD. Waist circumference also showed an independent association with the risk of SCA, but the association was a negative correlation with a 0.6% lower rate of SCA per 1 cm increase in waist circumference (Table 4).

## 4. Discussion

This study investigated the association of SCA with obesity, which is represented as BMI and waist circumference, based on a nationwide health insurance cohort of South Korea. Before consideration of the mediating risk factors, both general and central obesity were positively associated with risk of SCA. General obesity measured as BMI resulted in a J-shaped association with SCA, with highest SCA risk in the low body weight group (BMI < 18.5) followed by the obesity class II-III group (30.0 ≤ BMI), and the lowest SCA risk in the normal body weight group (18.5 ≤ BMI < 23). In contrast, central obesity measured as waist circumference reflected a linear association with SCA risk, resulting in a 2.6-fold increased risk of SCA in the highest waist circumference subgroup (100/95 cm ≤ waist circumference) compared with the reference group (waist circumference < 80/75 cm). However, the association between obesity and SCA risk was lost after adjustment of the covariates of metabolic disease and its surrogate marker (ɣ-GTP). In other words, a positive association of obesity and SCA was not present after adjustment of the covariates, which accentuates the mediating effect of metabolic disease and sociodemographic factors on SCA rather than the effect of obesity itself. Our study features discriminative strength through assessing the single exclusive nationwide health insurance system, comprising approximately 4.2 million participants, which is the largest population study assessing the association between obesity and SCA. Although the overall incidence of SCA was not high (<0.5%), sufficient cases of SCA (n = 16,352) were analyzed.

### 4.1. Obesity and SCA

Obesity is a well-established risk factor for mortality as well as atherosclerotic cardiovascular disease, which is represented as BMI, waist circumference, or waist-to-hip ratio. A J-shaped association between BMI and all-cause mortality is shown in prior studies [8,9]. Furthermore, central obesity assessed as waist circumference or waist-to-hip ratio reflected a robust association with mortality after adjustment of BMI, which led to a more comprehensive understanding of the linkage between obesity and mortality by evaluating both general obesity and central obesity [17]. However, both general and central obesity are strongly associated with various metabolic disorders such as hypertension, diabetes mellitus, dyslipidemia, and CKD, as shown in this study. It was unclear whether obesity itself or an associated metabolic disorder is the culprit risk factor for SCA.

Our study found a J-shaped association of SCA with BMI before adjustment of the covariates. Waist circumference also showed a linear association with SCA before covariate adjustment, which is a finding that is consistent with previous studies. However, the association of obesity and SCA was no longer present after adjustment for demographic factors (age and sex) and social habits, which was weakened even more after further consideration of the metabolic conditions and its surrogate marker, ɣ-GTP. Waist circumference showed a negative association with SCA after covariate adjustment (0.6% decreased risk of SCA per 1 cm increase in waist circumference: Table 4). Our findings suggest that obesity itself is not an independent risk factor for SCA, but is a surrogate marker of metabolic disorders and people demographics. Therefore, not only reducing body weight and waist circumference but also gaining a comprehensive understanding of the metabolic risk factors as a whole in each individual, and the successful management of metabolic disorders, may be important for primary prevention of SCA.

Obesity holds a strong correlation with metabolic disease as well as cardiovascular disease including coronary artery disease, heart failure, and cardiac arrhythmia. However, decreased risk of adverse cardiovascular outcomes in obese patients had been observed in various cardiovascular diseases, known as the obesity paradox [28,29]. The obesity paradox partly explains this loss of association of obesity and SCA. The exact mechanisms of the obesity paradox are yet to be established, but several hypotheses support the obesity paradox. Increased obesity alters hormonal and lipid mediators and cytokines—increased lipoproteins have a protective effect on inflammatory response, such as binding to endotoxin and increase of lymphocytes [30,31]. A decrease of adiponectin level and catecholamine response in obese patients also supports better clinical outcomes [32,33]. In addition, tumor necrosis factor-α I and II receptors produced by adipose tissue may promote an anti-arrhythmic environment, which may lead to decreased lethal arrhythmic events [34,35]. Furthermore, excessive fat and serum cholesterol in obese patients may serve as a reserve for acute inflammatory stress conditions that may provoke SCA [36]. Nonetheless, the use of BMI for measuring obesity should be interpreted with caution since BMI does not accurately reflect different components of body composition such as muscle mass and visceral fat. For instance, a previous study on cancer patients revealed that the obesity paradox was present when it was measured as BMI, but not in sarcopenic obesity patients [37]. It should also be acknowledged that most of the previous studies on the obesity paradox had focused on BMI as an assessment tool of obesity [38]. Therefore, to clarify the obesity paradox, further investigations that utilize more accurate methods to assess body composition and nutritional status are needed.

### 4.2. Prevention of SCA

Although the consequences of SCA events are highly dependent on the geographical accessibility of emergency medical services and the degree of training of citizens, the majority of SCA events impose considerable socioeconomic costs on their victims and family members [14]. In addition, even after the return of spontaneous circulation, neurologically complete recovery is challenging. Primary prevention of SCA with recognition of the underlying risk factors represents a key strategy in reducing the socioeconomic burden of SCA on public health.

Increased adiposity not only aggravates cardiovascular hemodynamics and leads to structural change of myocardium but also accelerates metabolic condition by the dysregulation of lipid metabolism, elevation of blood pressure, increased insulin resistance, and pro-inflammatory response [39,40]. Our study demonstrated a clear association between obesity (both by BMI and waist circumference) and various metabolic disorders. The increase of BMI and waist circumference was associated with an increased prevalence of metabolic disorders, including diabetes mellitus, hypertension, dyslipidemia, and chronic kidney disease. Clarifying whether obesity is independently associated with SCA or is indirectly associated with SCA through the influence of metabolic disorders is important for the prevention of SCA. If coexisting metabolic disorders are the culprit risk factor for SCA, weight reduction itself may not be sufficient to effectively prevent SCA, and concomitant management of metabolic disorders such as hypertension and diabetes may be more important.

### 4.3. Limitations

There are several limitations in this study. First, obesity defined through BMI and waist circumference was measured at the time of enrollment (2009), and temporal change of the parameters was not considered. Certain populations with systemic conditions such as malignancy or tuberculosis might have experienced acute change of body weight. The results of the current study cannot demonstrate cause and effect relationships. Consequently, further analysis of temporal change of BMI and waist circumference may provide more valuable clinical implications. In a similar vein, although obesity was quantified as BMI and waist circumference, it was not further analyzed specifically, such as analysis of fat mass and proportion of visceral fat. Second, since the outcome was restricted to out-of-hospital cardiac arrests, this study might have underestimated the actual incidence of SCA. Due to heterogenous etiologies of in-hospital cardiac arrests, it is difficult to distinguish the predisposing condition of in-hospital cardiac arrest by ICD-10 codes—whether it was a sudden, unexplained cardiac arrest or a hemodynamic collapse due to non-cardiac underlying conditions. Moreover, the major reason for excluding in-hospital cardiac arrest is due to its different clinical characteristics compared with out-of-hospital cardiac arrest [41]. Patients with in-hospital cardiac arrests are reported to have older age, higher proportion of non-shockable rhythm, and also a higher proportion of chronic illness such as infection, malignancy, or chronic respiratory disease [42]. Therefore, we restricted the analysis to out-of-hospital cardiac arrest to decrease heterogeneity and reduce other possible confounding factors. Last, our cohort exclusively consisted of an East Asian population and extrapolation to other ethnic groups should be undertaken with caution.

## 5. Conclusions

Obesity assessed as BMI and waist circumference did not show an independent association with SCA risk after adjustment of mediating risk factors. In conclusion, rather than focusing on obesity per se, an integrated approach with consideration of pre-existing metabolic disorders as well as people demographics and social habits might provide a better understanding and prevention of SCA.

## Figures and Tables

**Figure 1 jcm-12-02068-f001:**
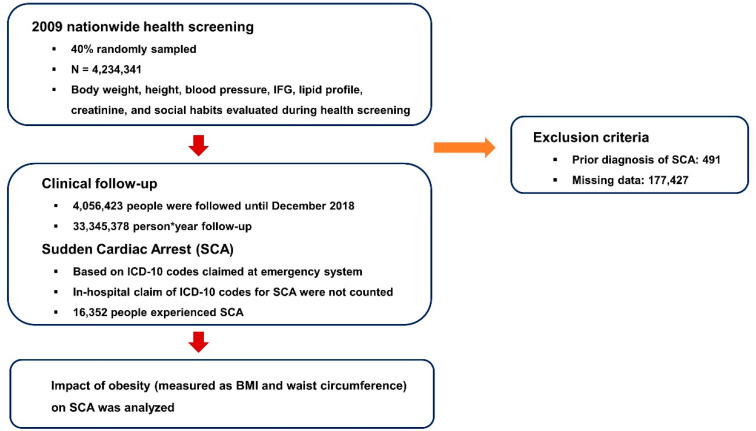
Study flow. BMI: body-mass index; ICD-10: International Classification of Disease, 10th revision; IFG: impaired fasting glucose; SCA: sudden cardiac arrest.

**Figure 2 jcm-12-02068-f002:**
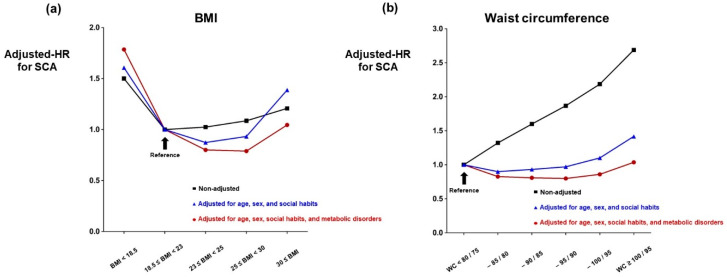
Impact of covariate adjustment. (**a**) The association between obesity (BMI ≥ 30) and SCA was lost when the impact of metabolic disorders (hypertension, diabetes mellitus, dyslipidemia, and CKD) was adjusted. (**b**) Waist circumference showed no association with SCA when various covariates including metabolic disorders were adjusted. BMI: body-mass index; CKD: chronic kidney disease; HR: hazard ratio; SCA: sudden cardiac arrest; WC: waist circumference. Waist circumference is expressed as centimeters. Social habits represent alcohol consumption, smoking status, regular exercise, and income. Metabolic disorders represent hypertension, diabetes mellitus, dyslipidemia, and CKD.

**Table 1 jcm-12-02068-t001:** Baseline demographics according to BMI.

	BMI					*p*-Value
	BMI < 18.5	18.5 ≤ BMI < 23	23 ≤ BMI < 25	25 ≤ BMI < 30	30 ≤ BMI	
	148,460	1,579,653	1,001,394	1,182,398	144,518	
Male	49,452 (33.3%)	750,152 (47.5%)	602,974 (60.2%)	749,493 (63.4%)	81,460 (56.4%)	<0.001
Age (years)	40.5 ± 16.6	45.2 ± 14.4	48.6 ± 13.4	49.1 ± 13.3	46.2 ± 13.9	<0.001
Age group						<0.001
20–29	52,975 (35.7%)	262,941 (16.7%)	85,034 (8.5%)	83,743 (7.1%)	16,923 (11.7%)	
30–39	33,026 (22.3%)	315,592 (20.0%)	175,039 (17.5%)	219,742 (18.6%)	35,411 (24.5%)	
40–49	23,966 (16.1%)	424,880 (26.9%)	272,812 (27.2%)	309,454 (26.2%)	34,747 (24.0%)	
50–59	13,701 (9.2%)	293,562 (18.6%)	241,153 (24.1%)	284,426 (24.1%)	28,106 (19.5%)	
60–69	10,819 (7.3%)	171,096 (10.8%)	151,983 (15.2%)	194,509 (16.5%)	20,081 (13.9%)	
70–79	10,593 (7.1%)	93,783 (5.9%)	66,893 (6.7%)	81,888 (6.9%)	8526 (5.9%)	
80–	3380 (2.3%)	17,799 (1.1%)	8480 (0.9%)	8636 (0.7%)	724 (0.5%)	
Waist circumference (cm)	66.2 ± 6.1	74.3 ± 6.8	81.2 ± 6.5	87.2 ± 6.5	96.8 ± 8.5	<0.001
Smoking						<0.001
Never-smoker	104,399 (70.3%)	1,009,981 (63.9%)	569,292 (56.9%)	641,966 (54.3%)	81,957 (56.7%)	
Ex-smoker	10,274 (6.9%)	176,612 (11.2%)	166,249 (16.6%)	211,763 (17.9%)	19,715 (13.6%)	
Current-smoker	33,787 (22.8%)	393,060 (24.9%)	265,853 (26.6%)	328,669 (27.8%)	42,846 (29.7%)	
Alcohol consumption						<0.001
Non-drinker	84,650 (57.0%)	844,228 (53.4%)	502,676 (50.2%)	580,170 (49.1%)	74,863 (51.8%)	
Mild-drinker	57,034 (38.4%)	635,906 (40.3%)	415,595 (41.5%)	483,904 (40.9%)	54,251 (37.5%)	
Heavy-drinker	6776 (4.6%)	99,519 (6.3%)	83,123 (8.3%)	118,324 (10.0%)	15,404 (10.7%)	
Regular exercise	14,083 (9.5%)	260,130 (16.5%)	201,750 (20.2%)	235,736 (19.9%)	25,058 (17.3%)	<0.001
Income (lowest 20%)	26,986 (18.2%)	285,175 (18.1%)	170,960 (17.1%)	198,438 (16.8%)	26,103 (18.1%)	<0.001
Diabetes mellitus	5032 (3.4%)	88,229 (5.6%)	89,877 (9.0%)	145,104 (12.3%)	25,156 (17.4%)	<0.001
Diabetes mellitus stage						<0.001
Non-diabetic	123,405 (83.1%)	1,201,431 (76.1%)	672,234 (67.1%)	710,909 (60.1%)	76,152 (52.7%)	
Impaired fasting glucose	20,023 (13.5%)	289,993 (18.4%)	239,283 (23.9%)	326,385 (27.6%)	43,210 (30.0%)	
New onset diabetes	2145 (1.4%)	30,453 (1.9%)	29,509 (3.0%)	49,482 (4.2%)	8994 (6.2%)	
Diabetic < 5 years	1339 (0.9%)	25,901 (1.6%)	29,589 (3.0%)	52,562 (4.5%)	10,096 (7.0%)	
Diabetic ≥ 5 years	1548 (1.0%)	31,875 (2.0%)	30,779 (3.1%)	43,060 (3.6%)	6066 (4.2%)	
Glucose (mg/dL)	90.9 ± 20.9	94.0 ± 21.3	97.9 ± 23.8	100.9 ± 25.7	104.9 ± 30.2	<0.001
Hypertension	14,751 (9.9%)	271,106 (17.2%)	280,148 (28.0%)	451,800 (38.2%)	73,908 (51.1%)	<0.001
Hypertension stage						<0.001
Non-hypertensive	91,041 (61.3%)	725,291 (45.9%)	307,165 (30.7%)	245,829 (20.8%)	16,651 (11.5%)	
Pre-hypertension	42,668 (28.7%)	583,256 (36.9%)	414,081 (41.4%)	484,769 (41.0%)	53,959 (37.3%)	
Hypertension	5325 (3.6%)	90,481 (5.7%)	84,145 (8.4%)	131,823 (11.2%)	24,305 (16.8%)	
Hypertension with medication	9426 (6.4%)	180,625 (11.4%)	196,003 (19.6%)	319,977 (27.1%)	49,603 (34.3%)	
Systolic blood pressure (mmHg)	113.7 ± 14.1	118.6 ± 14.4	123.4 ± 14.4	126.8 ± 14.5	131.2 ± 15.2	<0.001
Diastolic blood pressure (mmHg)	71.3 ± 9.3	74.0 ± 9.6	76.8 ± 9.7	79.0 ± 9.8	82.0 ± 10.5	<0.001
Dyslipidemia	8133 (5.5%)	187,002 (11.8%)	196,442 (19.6%)	300,990 (25.5%)	45,026 (31.2%)	<0.001
Dyslipidemia stage						<0.001
Total cholesterol < 240 (mg/dL)	140,327 (94.5%)	1,392,651 (88.2%)	804,952 (80.4%)	881,408 (74.5%)	99,492 (68.8%)	
Total cholesterol ≥ 240	4679 (3.2%)	95,287 (6.0%)	92,427 (9.2%)	136,280 (11.5%)	19,999 (13.8%)	
Total cholesterol ≥ 240 with medication	3454 (2.3%)	91,715 (5.8%)	104,015 (10.4%)	164,710 (13.9%)	25,027 (17.3%)	
Cholesterol (mg/dL)	177.8 ± 35.1	188.8 ± 39.1	197.9 ± 41.4	202.6 ± 42.1	206.3 ± 42.1	<0.001
High-density lipoprotein (mg/dL)	64.3 ± 37.7	59.7 ± 34.3	55.2 ± 30.4	53.0 ± 32.3	51.7 ± 29.6	<0.001
Low-density lipoprotein (mg/dL)	115.8 ± 351.8	119.1 ± 252.9	122.3 ± 181.8	123.3 ± 158.1	124.6 ± 152.8	<0.001
Chronic kidney disease	8198 (5.5%)	95,435 (6.0%)	71,342 (7.1%)	92,362 (7.8%)	11,257 (7.8%)	<0.001
eGFR (mL/min/1.73 m^2^)	93.1 ± 47.9	89.2 ± 44.7	86.5 ± 44.8	85.6 ± 44.6	87.0 ± 45.3	<0.001
ɣ-GTP *	18.7 (18.7–18.8)	21.3 (21.3–21.4)	27.4 (27.4–27.4)	33.8 (33.7–33.8)	40.5 (40.3–40.6)	<0.001

* expressed as median (interquartile range); BMI: body-mass index; eGFR: estimated glomerular filtration rate; ɣ-GTP: ɣ-glutamyl transferase.

**Table 2 jcm-12-02068-t002:** Baseline demographics according to waist circumference.

	Waist Circumference (Male/Female; cm)						*p*-Value
	WC < 80/75	80/75 ≤ WC < 85/80	85/80 ≤ WC < 90/85	90/85 ≤ WC < 95/90	95/90 ≤ WC < 100/95	100/95 ≤ WC	
	1,490,892	965,616	803,708	471,183	209,692	115,332	
Male	660,775 (44.3%)	598,889 (62.0%)	499,969 (62.2%)	293,667 (62.3%)	120,066 (57.3%)	60,165 (52.2%)	<0.001
Age (years)	42.4 ± 13.6	47.6 ± 13.2	50.2 ± 13.4	51.8 ± 13.6	52.7 ± 14.2	52.0 ± 15.3	<0.001
Age group							<0.001
20–29	318,255 (21.4%)	88,707 (9.2%)	49,348 (6.1%)	24,681 (5.2%)	11,448 (5.5%)	9177 (8.0%)	
30–39	335,158 (22.5%)	187,253 (19.4%)	133,262 (16.6%)	72,134 (15.3%)	31,459 (15.0%)	19,544 (17.0%)	
40–49	415,596 (27.9%)	272,408 (28.2%)	206,013 (25.6%)	108,027 (22.9%)	42,727 (20.4%)	21,088 (18.3%)	
50–59	242,128 (16.2%)	225,760 (23.4%)	202,887 (25.2%)	117,630 (25.0%)	48,855 (23.3%)	23,688 (20.5%)	
60–69	113,085 (7.6%)	128,347 (13.3%)	140,559 (17.5%)	95,715 (20.3%)	46,428 (22.1%)	24,354 (21.1%)	
70–79	55,993 (3.8%)	55,007 (5.7%)	63,160 (7.9%)	46,864 (10.0%)	25,367 (12.1%)	15,292 (13.3%)	
80–	10,677 (0.7%)	8134 (0.8%)	8479 (1.1%)	6132 (1.3%)	3408 (1.6%)	2189 (1.9%)	
Body mass index (kg/m^2^)	21.1 ± 2.1	23.5 ± 1.9	25.0 ± 2.0	26.5 ± 2.2	28.0 ± 2.4	30.7 ± 3.3	<0.001
Smoking							<0.001
Never-smoker	986,822 (66.2%)	532,393 (55.1%)	440,387 (54.8%)	257,592 (54.7%)	120,894 (57.7%)	69,507 (60.3%)	
Ex-smoker	148,357 (10.0%)	156,878 (16.3%)	142,207 (17.7%)	85,813 (18.2%)	35,049 (16.7%)	16,309 (14.1%)	
Current-smoker	355,713 (23.9%)	276,345 (28.6%)	221,114 (27.5%)	127,778 (27.1%)	53,749 (25.6%)	29,516 (25.6%)	
Alcohol consumption							<0.001
Non-drinker	793,738 (53.2%)	470,369 (48.7%)	402,489 (50.1%)	240,337 (51.0%)	113,560 (54.2%)	66,094 (57.3%)	
Mild-drinker	612,902 (41.1%)	412,931 (42.8%)	326,697 (40.7%)	182,237 (38.7%)	74,397 (35.5%)	37,526 (32.5%)	
Heavy-drinker	84,252 (5.7%)	82,316 (8.5%)	74,522 (9.3%)	48,609 (10.3%)	21,735 (10.4%)	11,712 (10.2%)	
Regular exercise	245,521 (16.5%)	190,060 (19.7%)	156,820 (19.5%)	88,629 (18.8%)	37,125 (17.7%)	18,602 (16.1%)	<0.001
Income (lowest 20%)	276,206 (18.5%)	162,310 (16.8%)	133,818 (16.7%)	78,807 (16.7%)	36,187 (17.3%)	20,334 (17.6%)	<0.001
Diabetes mellitus	54,778 (3.7%)	75,069 (7.8%)	89,473 (11.1%)	69,195 (14.7%)	38,400 (18.3%)	26,483 (23.0%)	<0.001
Diabetes mellitus stage							<0.001
Non-diabetic	1,183,986 (79.4%)	665,077 (68.9%)	502,496 (62.5%)	268,100 (56.9%)	109,731 (52.3%)	54,741 (47.5%)	
Impaired fasting glucose	252,128 (16.9%)	225,470 (23.4%)	211,739 (26.4%)	133,888 (28.4%)	61,561 (29.4%)	34,108 (29.6%)	
New onset diabetes	22,574 (1.5%)	27,518 (2.9%)	30,030 (3.7%)	21,671 (4.6%)	11,218 (5.4%)	7572 (6.6%)	
Diabetic < 5 years	15,676 (1.1%)	23,454 (2.4%)	30,162 (3.8%)	24,940 (5.3%)	14,740 (7.0%)	10,515 (9.1%)	
Diabetic ≥ 5 years	16,528 (1.1%)	24,097 (2.5%)	29,281 (3.6%)	22,584 (4.8%)	12,442 (5.9%)	8396 (7.3%)	
Glucose (mg/dL)	92.3 ± 18.7	97.1 ± 23.1	100.0 ± 25.6	102.5 ± 27.5	104.8 ± 29.8	108.0 ± 33.4	<0.001
Hypertension	195,526 (13.1%)	240,734 (24.9%)	276,994 (34.5%)	203,635 (43.2%)	107,014 (51.0%)	67,810 (58.8%)	<0.001
Hypertension stage							<0.001
Non-hypertensive	746,795 (50.1%)	314,141 (32.5%)	196,915 (24.5%)	86,969 (18.5%)	29,483 (14.1%)	11,674 (10.1%)	
Pre-hypertension	548,571 (36.8%)	410,741 (42.5%)	329,799 (41.0%)	180,579 (38.3%)	73,195 (34.9%)	35,848 (31.1%)	
Hypertension	75,002 (5.0%)	80,335 (8.3%)	81,614 (10.2%)	54,905 (11.7%)	26,803 (12.8%)	17,420 (15.1%)	
Hypertension with medication	120,524 (8.1%)	160,399 (16.6%)	195,380 (24.3%)	148,730 (31.6%)	80,211 (38.3%)	50,390 (43.7%)	
Systolic blood pressure (mmHg)	117.3 ± 13.9	122.7 ± 14.3	125.4 ± 14.5	127.6 ± 14.8	129.5 ± 15.1	131.8 ± 15.7	<0.001
Diastolic blood pressure (mmHg)	73.3 ± 9.5	76.5 ± 9.7	78.1 ± 9.8	79.3 ± 10.0	80.3 ± 10.2	81.7 ± 10.7	<0.001
Dyslipidemia	142,044 (9.5%)	170,261 (17.6%)	187,884 (23.4%)	130,610 (27.7%)	66,440 (31.7%)	40,354 (35.0%)	<0.001
Dyslipidemia stage							<0.001
Total cholesterol < 240 (mg/dL)	1,348,848 (90.5%)	795,355 (82.4%)	615,824 (76.6%)	340,573 (72.3%)	143,252 (68.3%)	74,978 (65.0%)	
Total cholesterol ≥ 240	80,387 (5.4%)	85,808 (8.9%)	85,795 (10.7%)	54,982 (11.7%)	26,340 (12.6%)	15,360 (13.3%)	
Total cholesterol ≥ 240 with medication	61,657 (4.1%)	84,453 (8.8%)	102,089 (12.7%)	75,628 (16.1%)	40,100 (19.1%)	24,994 (21.7%)	
Cholesterol (mg/dL)	186.6 ± 37.4	196.6 ± 40.8	201.1 ± 42.8	203.2 ± 43.3	204.9 ± 43.3	206.2 ± 44.2	<0.001
High-density lipoprotein (mg/dL)	60.6 ± 33.7	55.8 ± 32.6	53.9 ± 31.9	52.6 ± 31.9	52.3 ± 31.9	52.2 ± 32.1	<0.001
Low-density lipoprotein (mg/dL)	120.3 ± 292.0	120.6 ± 167.8	122.0 ± 142.0	122.5 ± 143.4	122.8 ± 132.4	123.2 ± 117.3	<0.001
Chronic kidney disease	77,727 (5.2%)	65,384 (6.8%)	62,037 (7.7%)	40,381 (8.6%)	20,738 (9.9%)	12,327 (10.7%)	< 0.001
eGFR (mL/min/1.73 m^2^)	90.1 ± 45.0	86.9 ± 43.9	86.0 ± 46.3	85.2 ± 44.6	84.7 ± 43.4	85.4 ± 44.8	<0.001
ɣ-GTP *	20.1 (20.1–20.1)	27.1 (27.1–27.2)	31.2 (31.1–31.2)	34.8 (34.7–34.9)	36.6 (36.5–36.7)	38.9 (38.7–39.0)	<0.001

* expressed as median (interquartile range); WC: waist circumference; BMI: body-mass index; eGFR: estimated glomerular filtration rate; ɣ-GTP: ɣ-glutamyl transferase.

**Table 3 jcm-12-02068-t003:** Impact of BMI and waist circumference on SCA.

	n	SCA	Follow-Up Duration (Person-Years)	Incidence	Hazard Ratio with 95% Confidence Interval					
					Univariate	Multivariate 1	Multivariate 2	Multivariate 3	Multivariate 4	Multivariate 5
**BMI**										
BMI < 18.5	148,460	830	1,196,986	0.69	1.50 (1.40–1.61)	1.70 (1.58–1.83)	1.61 (1.49–1.73)	1.78 (1.65–1.91)	1.79 (1.66–1.92)	1.79 (1.66–1.92)
18.5 ≤ BMI < 23	1,579,653	6016	12,966,752	0.46	1 (reference)	1 (reference)	1 (reference)	1 (reference)	1 (reference)	1 (reference)
23 ≤ BMI < 25	1,001,394	3924	8,249,250	0.48	1.02 (0.98–1.07)	0.85 (0.81–0.88)	0.87 (0.84–0.91)	0.80 (0.77–0.84)	0.80 (0.77–0.83)	0.78 (0.75–0.81)
25 ≤ BMI < 30	1,182,398	4915	9,743,125	0.50	1.09 (1.05–1.13)	0.90 (0.86–0.93)	0.93 (0.90–0.97)	0.80 (0.76–0.83)	0.79 (0.76–0.82)	0.74 (0.71–0.77)
30 ≤ BMI	144,518	667	1,189,264	0.56	1.21 (1.12–1.31)	1.36 (1.25–1.47)	1.39 (1.28–1.50)	1.06 (0.97–1.14)	1.05 (0.96–1.13)	0.94 (0.87–1.02)
**Waist circumference (male/female; cm)**										
<80/75	1,490,892	4286	12,283,975	0.35	1 (reference)	1 (reference)	1 (reference)	1 (reference)	1 (reference)	1 (reference)
–85/80	965,616	3665	7,945,666	0.46	1.32 (1.26–1.38)	0.88 (0.84–0.92)	0.90 (0.86–0.94)	0.83 (0.79–0.87)	0.83 (0.79–0.87)	0.80 (0.76–0.83)
–90/85	803,708	3690	6,603,475	0.56	1.60 (1.53–1.67)	0.91 (0.87–0.95)	0.93 (0.89–0.97)	0.81 (0.78–0.85)	0.81 (0.77–0.85)	0.76 (0.73–0.80)
–95/90	471,183	2520	3,859,942	0.65	1.87 (1.78–1.96)	0.94 (0.90–0.99)	0.97 (0.92–1.02)	0.81 (0.77–0.85)	0.80 (0.76–0.84)	0.74 (0.70–0.77)
–100/95	209,692	1310	1,714,019	0.76	2.19 (2.06–2.33)	1.07 (1.00–1.14)	1.10 (1.03–1.17)	0.87 (0.82–0.93)	0.86 (0.81–0.92)	0.78 (0.73–0.83)
≥100/95	115,332	881	938,301	0.94	2.69 (2.50–2.89)	1.39 (1.29–1.50)	1.42 (1.32–1.53)	1.05 (0.98–1.14)	1.04 (0.96–1.12)	0.92 (0.86–0.99)

Incidence is per 1000 person-years follow-up. BMI: body-mass-index; SCA: sudden cardiac arrest; ɣ-GTP: gamma-glutamyl transferase. Multivariate model 1: adjusted for age and sex. Multivariate model 2: adjusted for model 1 plus smoking, alcohol, regular exercise, and income. Multivariate model 3: adjusted for model 2 plus hypertension, diabetes mellitus, and dyslipidemia. Multivariate model 4: adjusted for model 2 plus hypertension, diabetes mellitus, dyslipidemia, and chronic kidney disease. Multivariate model 5: adjusted for model 4 plus ɣ-GTP.

**Table 4 jcm-12-02068-t004:** Multivariate model for SCA prediction.

	Hazard Ratio with 95% Confidence Interval	*p*-Value
Age (year)	1.08 (1.08–1.08)	<0.001
Sex		<0.001
Male	2.35 (2.25–2.46)	
Female	1 (reference)	
Smoking status		<0.001
Non-smoker	1 (reference)	
Ex-smoker	1.14 (1.09–1.20)	
Current-smoker	1.81 (1.74–1.89)	
Alcohol consumption		<0.001
Non-drinker	1 (reference)	
Mild-drinker	0.77 (0.74–0.80)	
Heavy-drinker	0.75 (0.71–0.79)	
Regular exercise		<0.001
No	1 (reference)	
Yes	0.89 (0.86–0.93)	
Income		<0.001
High	1 (reference)	
Low	1.09 (1.05–1.14)	
Hypertension		<0.001
Non-hypertension	1 (reference)	
Pre-hypertension	1.16 (1.11–1.21)	
Hypertension	1.51 (1.44–1.59)	
Diabetes mellitus		<0.001
Non-DM	1 (reference)	
IFG	1.06 (1.02–1.10)	
DM	1.74 (1.67–1.81)	
Dyslipidemia		<0.001
Total cholesterol < 240 (mg/dL)	1 (reference)	
Total cholesterol ≥ 240	1.09 (1.03–1.15)	
Total cholesterol ≥ 240 with medication	0.97 (0.93–1.01)	
Chronic kidney disease	1.47 (1.41–1.53)	<0.001
Waist circumference (cm; continuous)	0.99 (0.99–1.00)	<0.001
ɣ-GTP (unit; continuous)	1.00 (1.00–1.00)	<0.001

DM: diabetes mellitus; ɣ-GTP: gamma-glutamyl transferase; IFG: impaired fasting glucose; SCA: sudden cardiac arrest.

## Data Availability

The data underlying this article are available in the article.

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
