# Peer review of "Obesity Is Indirectly Associated with Sudden Cardiac Arrest through Various Risk Factors"

_jcm, 2023, doi:10.3390/jcm12052068_

Round 1
Reviewer 1 Report
Review of article: “Obesity is indirectly associated with sudden cardiac arrest through various risk factors
I read the interesting article written by Kim YG and coll. In this article the authors analyzed retrospectively data concerning obesity in a large amount of Korean people. The authors take into account body mass index and waist circumference. The main aim was to search a correlation between obesity related data and sudden cardiac arrest. The authors collected data derived from national database and obtained from regular check-up that Korean people regularly underwent. A great amount of people, over 4 milion, is enrolled in the article. The main results is that only an indirect correlation is shown from data analysis. The topic is very important and based on a great number of patients. Although these strong points some important consideration must be underlined with the aim to improve the article:
- is the check up free or patients must pay it? In a previous section authors referred that all the Korean national system is free but it is not explained in the line 80;
- in hospital cardiac arrest are excluded from the study, as said in method section; it is not clear the motivation for this decision (line 108);
- old collected data; more than 10 years (from 2009); authors explained this as a limitation but this could limit results concerning obesity;
- it is not explained the type of study (retrospective as I suppose or prospective);
- the primary outcome is indicated to be the incidence of SCA during the follow up; in my opinion the topic is correlation among obesity and SCA and authors have to clarify this point;
- in statistic section the authors don’t indicate how data are presented (mean, median….standard deviation or range..);
- figure 1 representing the study flow is not clear; people experienced SCA are a subgroup of people followed until December 2018 while in the picture these patients seem to come from the first group;
- in result section it is not reported the correct number of people analyzed in table 1 and 2. In fact 4,056,423 people are analyzed in table 1 and 2 and not 4,234,341 as expressed in result; do they study only patients after follow-up (2018) or people during check-up (2009)?
- in limitations the authors indicate that the third limitation is the lack of data concerning neurological outcome; this is not an end point of the study;
- Grammar mistakes: line 26 were instead of was, line 33 diseases instead of disease, line 50 imposes, line 65 represents instead of represent, line 146 were instead of was referring to definitions.
I hope these points may help authors to improve their study on a important topic, opening new idea and ways of study.
Regards
Reviewer 2 Report
Dear Authors, thank you for the opportunity to review your manuscript. The idea of this analysis is very interesting, however I have few questions:
Line 50 – Please use already introduced abbreviations
Line 110 – Why GE bleeding was used as the exclusion?
Line 114 – Please describe the technique and accuracy of WC measurement
Line 129 – Why 100 cigarettes were used for the cut-off?
Line 155 – Model 3 includes the main components of metabolic syndrome, but renal function is not one of the. I suggest an additional model with analyzes only classical components of metabolic syndrome
High BMI (obesity) is the basis of metabolic syndrome. I suggest an additional comparison between the patients who had obesity without accompanying complications (without metabolic syndrome) and between the ones with the components of metabolic syndrome (hypertension, diabetes, dyslipidemia) but without obesity, to support the presented results.
Please provide the basis on which you divided BMI 18,5-25 into normal and high normal body weight, as WHO criteria provide a normal range of 18,5-25.
Why the cut-off for dyslipidemia was TC >240, while the value for diagnosis is lower?
Line 297 – There are also obesity paradox hypotheses related to body composition, I suggest mentioning them.
Line 308 – Please provide the calculation for this correlation
Line 320 – the Authors underline that this study does not prove a cause/result relationship which is correct due to the type of this study, while the conclusions presented in the discussion suggest this cause/result relationship. I suggest more cautious conclusions.
Round 2
Reviewer 1 Report
Dear the authors modify the article in a satifing way: in particular.
1. explanation for the reason to exclude the in hospital cardiac arrest
2 . the authors indicated the type of study
3 improving in statistic section;
4 better the flow diagram;
Author Response
Thank you very much for your valuable feedback.
Reviewer 2 Report
Dear Authors,
Thank you very much for your revisions. I believe that the manuscript improved its quality. However, during the revision, you uploaded the file prepared for another journal, not JCM. Please provide the manuscript which meets the formatting criteria of MDPI by using the template.
There are also a few punctuations and grammar errors that need corrections.
Author Response
Thank you very much for your valuable feedback.
The manuscript is formated by JCM assistant editor now, and minor grammar mistakes will be fixed by english editing team.